

# LINC00969 inhibits proliferation with metastasis of breast cancer by regulating phosphorylation of PI3K/AKT and ILP2 expression through HOXD8

Xiaoyun Wen[1], Ya Hou[2], Liang Zhou[2] and Xiansong Fang[3]

[1] Clinical Laboratory, The First Affiliated Hospital of Gannan Medical University, Ganzhou, China
[2] The First School of Clinical Medicine,Gannan Medical University, Ganzhou, China
[3] Blood transfusion department,The First Affiliated Hospital of Gannan Medical University, Ganzhou, China

## ABSTRACT

**Background.** Breast cancer (BC) is a malignancy that is inadequately treated and poses a significant global health threat to females. The aberrant expression of long noncoding RNAs (lncRNAs) acts as a complex with a precise regulatory role in BC progression. LINC00969 has been linked to pyroptotic cell death and resistance to gefitinib in lung cancer cells. However, the precise function and regulatory mechanisms of LINC00969 in BC remain largely unexplored.

**Methods.** Cell proliferation, migration, and invasion of BC cells were evaluated using CCK-8 and Transwell assays. Western blotting was employed to analyze the protein expression levels of HOXD8, ILP2, PI3K, t-AKT, and p-AKT.

**Results.** LINC00969 was drastically reduced in BC tissues LINC00969 overexpression markedly suppressed proliferation, migration, and invasion, and blocked PI3K and p-AKT protein expression in MCF-7 cells. Activation of the PI3K/AKT pathway reversed the suppressive effect of LINC0096 overexpression on the proliferation, migration, and invasion of MCF-7 cells. Moreover, LINC00969 overexpression enhanced HOXD8 and blocked ILP2 protein expression in MCF-7 cells. In contrast, activating the PI3K/AKT pathway had no effect on HOXD8 and blocked ILP2 protein expression in MCF-7 cells overexpressing LINC00969. HOXD8 knockdown enhanced ILP2, PI3K, and p-AKT protein expression, and the proliferation, migration, and invasion of MCF-7 cells co-transfected with si-HOXD8 and ov-LINC00969. LINC00969 regulated HOXD8 *via* binding to miR-425-5p.

**Conclusion.** LINC00969 inhibits the proliferation and metastasis of BC cells by regulating PI3K/AKT phosphorylation through HOXD8/ILP2.

Corresponding author
Xiansong Fang, fangxiansong2020@163.com

## INTRODUCTION

Breast cancer (BC) is an untreated malignancy that poses a serious threat to females' health worldwide (*Siegel et al., 2023*). The incidence of BC has been increasing recently. Epidemiological statistics from 2020 indicated that approximately 420,000 new cases of BC were reported in Chinese females (*Giaquinto et al., 2022*). Over the last decade, many

advances have been made in the diagnosis of BC. However, the overall prognosis of BC remains unsatisfactory because most patients are initially diagnosed at an advanced stage, accompanied by metastasis or recurrence (*Li, Jin & Li, 2021*; *Wilcken, 2023*). The molecular mechanism underlying BC proliferation and metastasis is a complex biological process and involve the activation or silencing of genes with epigenetic modifications (*Barzaman et al., 2020*). The specific molecular mechanism remains largely unknown. Therefore, the search for novel molecular markers and therapeutic targets is vital for prolonging patient survival.

Recently, researchers have confirmed that the aberrant expression of long noncoding RNAs (lncRNAs) acts the complex regulatory roles in cancer progression, including BC (*Esposito et al., 2019*; *Yousefi et al., 2020*). LncRNA, which had high tissue and cell type specificity, can function as the diagnostic and prognostic biomarker and regulate the malignant function and multidrug resistance of BC cells, making them potential therapeutic target (*Jin et al., 2021*; *Li et al., 2023*). However, the role of certain lncRNAs in BC is still unclear. Therefore, this study selected lncRNA for further research. LINC00969, located on human chromosome 3q29, is aberrantly expressed and accelerates intervertebral disc recession (*Yu et al., 2019*). LINC00969 is associated with lung cancer cell pyroptosis and gefitinib resistance, suggesting a pivotal oncogene role for LINC00969 in cancer (*Dai et al., 2023*). However, the role of LINC00969 in BC remains unclear.

The PI3K/AKT pathway (PA-P) plays a crucial role in various malignant functions of cancer cells, including growth, metastasis, and angiogenesis. The PA-P has been identified as a vital facilitation pathway in *Miricescu et al. (2020)*. Within PA-P, membrane receptor binding and cytokine binding, such as IGF-1, activate PI3K subunit p85, which recruits PI3K subunit p110. This leads to the phosphorylation of PIP2 on the inner surface of the membrane, generating PI3P. PI3P serves as the second messenger to further activate AKT, facilitating downstream signaling events (*Bertucci, Bertucci & Gonçalves, 2023*). Targeting the PA-P with natural or synthetic drugs shows promise for cancer treatment (*Cerma et al., 2023*; *Yuan et al., 2023*). Notably, lncRNA can regulate the malignant function of BC cell *via* PA-P (*Maharati & Moghbeli, 2023*). For example, lncRNA-BC069792 is expressed at low levels in BC and suppresses BC cell proliferation and metastasis by inhibiting phosphorylated protein kinase B (p-AKT) (*Zhang et al., 2023*). However, the relationship between LINC00969 and PA-P in BC remains unclear.

In the present study, LINC00969 expression was evaluated in BC samples. Subsequently, we assessed its biological function and investigated the underlying regulatory mechanisms in BC cells.

## MATERIALS & METHODS

### Bioinformatics analyses

The Gene Expression Profiling Interactive Analysis (GEPIA) database was used to assess LINC00969 (Gene ID: 440993) expression in BC and normal breast tissues.

### Cell culture and transfection

Four BC cell lines (MCF-7, MDA-MB-231, SUM190PT, and SK-BR-3) and the immortalized breast cell line MCF-10A were acquired from the Shanghai Cell Bank of

Academy (Shanghai, China) and cultured in RPMI-1640 medium (Gibco, Franklin Lakes, NJ, USA) supplemented with 10% fetal bovine serum (FBS; Gibco) in a humidified incubator in an atmosphere of 5% CO2 at 37 °C. LINC00969 overexpression (ov-LINC00969), empty pcDNA3.1 plasmids (ov-NC), small interfering RNA (siRNA) against HOXD8 (si-HOXD8), and siRNA negative control (si-NC) were purchased from GenePharma (Shanghai, China). Cells ($1 \times 10^6$) were transfected using the Lipofectamine 2000 reagent (Invitrogen, Carlsbad, CA, USA) according to the manufacturer's instructions. To study the function of LINC00969, MCF-7 cells were transfected with ov-NC and ov-LINC00969 plasmids, while cells without transfection plasmids served as control groups (cell groups). To investigate the relationship between LINC00969 and HOXD8, MCF-7 cells were transfected with ov-LINC00969, ov-LINC00969+si-NC, and ov-LINC00969+si-HOXD8. To investigate the relationship between LINC00969 and PA-P, MCF-7 cells were transfected with LINC00969 and treated with nothing (ov-LINC00969 group), PBS (ov-LINC00969+PBS group), and PA-P activator IGF-1 (100 ng/mL; Sigma-Aldrich, St. Louis, MO, USA; ov-LINC00969+IGF-1 group). si-HOXD8 sequences: 5′-CTCTAGAGTTGGAAAAGGAAT- 3′ and si-NC sequences: 5′-TTCTCCGAACGTGTCACGTTT-3′.

## Quantitative reverse transcription-polymerase chain reaction (qRT-PCR)

LINC00969 expression in the BC and normal cell lines, ov-LINC00969 transfected MCF-7 cells, and ov-NC transfected MCF-7 cells was measured by qRT-PCR. The cells ($1 \times 106$) were collected immediately by centrifugation using a model JIDI-5R centrifuge (Guangzhou JiDi Instrument Co., Ltd., Guangzhou, China) for testing after culturing according to the experimental conditions. Total RNA from the cells was separated using TRIzol reagent (Invitrogen) on ice. RNase-free DNase I (Promega, Beijing, China) was used to digest the genome with purity (A260/A280) at 1.8–2.0 and yield close to 40 μg. Then, RNA was stored in −80° refrigerator. RNA integrity was detected by agarose gel electrophoresis, producing clear 28S and 18S rRNA bands, and RNA integrity number $\geq 7$. RNA concentration was determined using a BioPhotometer Plus spectrophotometer (Eppendorf, Hamburg, Germany). A total of 1.0 μg RNA was converted to cDNA by applying oligo (dT) in a total volume of 20 μL/5μg in a reverse transcription system (Promega, Madison, WI, USA) with 20 ng/μL of reverse transcriptase concentrations. Amplicon length is 100 bp and location of amplicon: TGGTG-GAAGTGAGACCGAAATAGACTCTGAAGTAATATCTGGACTTTGAAATTGCAGCT-GCTCCCAACCCCAGAACCTTTGCTGTCTTGGGACGGATCAC, and compare using NCBI database. The amplification product on the exon of 2.3 is the target splicing variant. PCR was performed using an ABI PRISM® 7500 Sequence Detection System (Applied Biosystems, Waltham, MA, USA) with 10 μL 2x SYBR GREEN qPCR Super Mix (Invitrogen) including polymerase using PCR tubes (AXYGEN, PCR-0208-C). The PCR conditions were 40 cycles of 50 °C for 2 min, 95 °C for 15 s, 60 °C for 32 s. Melting curve analysis: temperature 60 °C–95 °C. The sequences of primers were: LINC00969 forward (2 exon): 5′-TGGTGGAAGTGAGACCGAAAT-3′,

reverse (3 exon): 5′-GTGATCCGTCCCAAGACAGC-3′; glyceraldehyde 3-phosphate dehydrogenase (GAPDH) forward: 5′-GAAGGTGAAGGTCGGAGTC-3′, and reverse: 5′-GAAGATGGTGATGGGATTTC-3′. Fold changes in RNA abundances were computed applying the $2 - \Delta\Delta CT$ method with GAPDH as the internal reference for LINC00969, the primers were used at a concentration of 10 μmol/L (0.5 μL), and slope fluctuates between $-3.59$ and 3.1, $R2 \geq 0.9$. All experiments were independently repeated three times.

## Cell growth with proliferation assay

Cell growth and proliferation were evaluated using the CCK-8 assay. In brief, transfected cells were dispensed in wells of 96-well plates (each sample in triplicate), and incubated in a humidified 5% $CO2$ incubator at 37 °C for 0, 24, 48, and 72 h. At each time point, CCK-8 solution (10 μL) was added to every well and incubated for 2 h at 37 °C. A microplate reader (multiscan MK3, Thermo Fisher Scientific, Waltham, MA, USA) was used to measure the absorbance of cells at 450 nm. In addition, after culture 48 h, the cells were photographed by microscopy (OLYMPUS CKX41, U-CTR30-2; Olympus, Tokyo, Japan).

## Cell apoptosis and cycle assay

Apoptosis was detected using the Annexin V-FITC Cell Apoptosis Detection Kit (Keygen, Nanjing, China), and cell cycle was detected using the Cell Cycle Detection Kit (Keygen). Cell apoptosis and cycle were analyzed by Flow cytometry (BD, San Diego, CA, USA).

## Transwell assay of cell migration and invasion

Transwell chambers (Corning, Corning, NY, USA) were used to analyze cell migration and invasion. The chambers were pre-coated with Matrigel (BD, Santa Clara, CA, USA) to assay invasion or were not coated to assay migration. Transfected cells were added to the top chamber with or without Matrigel in the FBS-free medium. Each lower chamber contained 10% FBS. After 48 h incubation at 37 °C with 5% $CO_2$, cells were exposed to methanol. Cells that had not migrated or invaded remained in the tope chamber. These cells were wiped off the membrane surface in the top chamber using a cotton swab. The bottom surface of each membrane was stained with crystal violet. Five regions were randomly selected, and the cells were counted by microscopy (OLYMPUS CKX41).

## Western blotting

RIPA lysis buffer containing protease inhibitor (Beyotime, Shanghai, China) was used to extract total protein. Denatured total protein (30 μg) was separated by SDS-PAGE and then transferred to PVDF membranes (Millipore, Billerica, MA, USA). After stabilizing, the membranes were soaked at 4 °C overnight with diluted primary antibodies (all from Abcam, Cambridge, UK) to HOXD8 (ab229321, 1:1000), inhibitor of apoptosis-like protein-2 (ILP2; ab9664, 1:1000), phosphoinositide 3-kinase (PI3K; ab191606, 1:1500), p-AKT (ab38449, 1:800), t-AKT (total AKT; ab8805, 1:2000), or GAPDH (ab9485, 1:3000). After rinsing, the membranes were incubated with horseradish peroxidase-labeled secondary antibody (1:2000, Abcam) for 2 h at 25 °C and then washed. Finally, the protein bands were detected by enhanced chemiluminescence detection kit (Beyotime) and ChemiDoc™ XRS system (Bio-Rad, Hercules, CA, USA). Assay carried out by core lab.

### Luciferase experiment assay

The miRNAs that binds to HOXD8 3′UTR and LINC00969 were analyzed by Targetscan 8.0 and LncBase 2.0, respectively. Wild type (WT) and mutant (mut) sequences of LINC00969 were inserted to psi-CHECK2. Similarly, WT HOXD8 3′UTR and mut HOXD8 3′UTR sequences were also inserted to psi-CHECK2. Then all psi-CHECK2 were transfected into MCF7, respectively. There transfection MCF7 were co-transfected miR-425-5p mimic and NC mimic, respectively. After 24 h, the luciferase activity was assessed using the Dual-Light Chemiluminescent Reporter Gene Assay System (Applied Biosystems, Foster City, USA) and Renilla/firefly (R/F) luciferase activity rate were calculated.

### Statistical analysis

Statistical analysis was performed using SPSS version 22.0 (IBM, Armonk, NY, USA). Data are presented as means $\pm$ standard deviation (SD). A one-way ANOVA was performed to detect significant differences between the three groups. $P$-values $< 0.05$ were considered significant.

## RESULTS

### LINC00969 is markedly decreased in BC

To study LINC00969's role in BC, LINC00969 expression was analyzed. The GEPIA online database analysis data indicated that LINC00969 expression was markedly decreased in BC tissues compared to normal tissues (Fig. 1A). Next, LINC00969 expression in BC and normal cell lines was assessed by qRT-PCR. LINC00969 expression in four BC cell lines dramatically decreased, especially in MCF-7 cells, compared to that in MCF-10A cells (Fig. 1B). Therefore, MCF-7 cells were selected for further studies.

### LINC00969 overexpression suppresses proliferation, cycle, migration, and invasion and promotes apoptosis in BC cells

To explore the biological function of LINC00969 in BC cells, ov-LINC00969 plasmid was transfected to upregulate LINC00969 expression. qRT-PCR analysis revealed the markedly upregulated expression of LINC00969 in MCF-7 cells in ov-LINC00969 group (Fig. 2A). Functionally, the microscope images showed the obviously decrease in the number of cells in the ov-LINC00969 group (Fig. S1A). CCK8 assay results showed that LINC00969 overexpression significantly suppressed MCF-7 cell proliferation at 48 and 72 h compared with the ov-NC-transfected cells (Fig. 2B). In addition, LINC00969 overexpression significantly promoted apoptosis and enhanced G1 phage accumulation compared with the ov-NC-transfected cells (Figs. S1B and S1C). Transwell assay results demonstrated that LINC00969 overexpression remarkably depressed MCF-7 cell migration and invasion abilities compared to ov-NC-transfected cells (Figs. 2C and 2D).

### LINC00969 overexpression suppresses BC cells malignant function through PA-P

Therefore, we investigated whether PA-P is regulated by LINC000969. As expected, western blot assays showed that LINC000969 overexpression markedly decreased PI3K and p-AKT expression in MCF-7 cells compared to ov-NC-transfected cells (Fig. 3A).

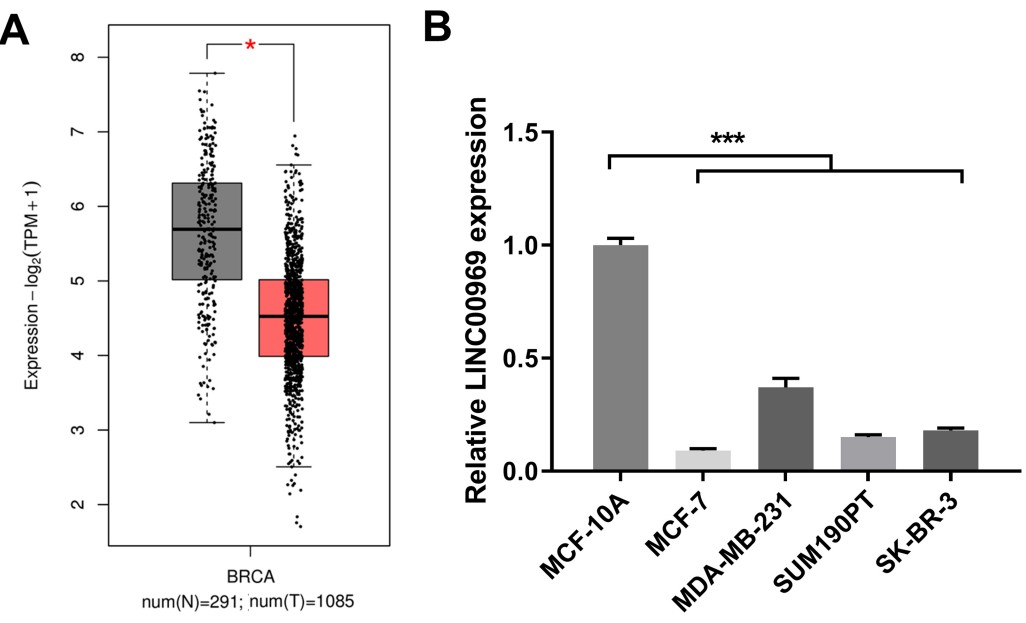

**Figure 1** **LINC00969 expression is decreased in BC.** (A) LINC00969 expression in BC tissues was analyzed using the GEPIA online database. (B) qRT-PCR was used to evaluate the LINC000969 expression of BC cells. (*$P < 0.05$, ***$P < 0.001$).

Next, we examined the role of PA-P in BC cells overexpressing LINC000969. MCF-7 cells overexpressing LINC00969 were treated with IGF-1 to activated PA-P. PI3K and p-AKT protein expression was markedly increased in MCF-7 cells overexpressing LINC00969 (Fig. 3B). We further investigated whether activating PA-P in MCF-7 cells overexpressing LINC00969 could reverse the effects of LINC00969 overexpression in MCF-7 cells. Results of the CCK-8 and Transwell assays showed that activating PA-P could remarkably enhance proliferation, migration, and invasion abilities of MCF-7 cells accompanied by LINC00969 overexpression (Figs. 3C–3E).

### HOXD8 expression is regulated by LINC00969 in BC

Previous studies confirmed that HOXD8 is closely associated with BC cell proliferation, migration, and invasion (*Zhang et al., 2021*; *Wen, Chen & Fang, 2021*). Western blotting revealed that HOXD8 protein expression in four BC cell lines was dramatically decreased, especially in MCF-7 cells, compared with that in MCF-10A cells (Fig. 4A). Moreover, LINC00969 overexpression enhanced HOXD8 protein expression in MCF-7 cells (Fig. 4B), whereas the activation of PA-P in MCF-7 cells accompanied by LINC00969 overexpression had no effect on HOXD8 protein expression, indicating that HOXD8 may be upstream of PA-P (Fig. 4C).

### HOXD8 knockdown alleviates the repressive effect of LINC00969 overexpression on BC cells *via* PA-P

To further investigate the relationship between LINC00969 and HOXD8, MCF-7 cells transfected si-HOXD8 were also transfected with ov-LINC00969. Western blotting results

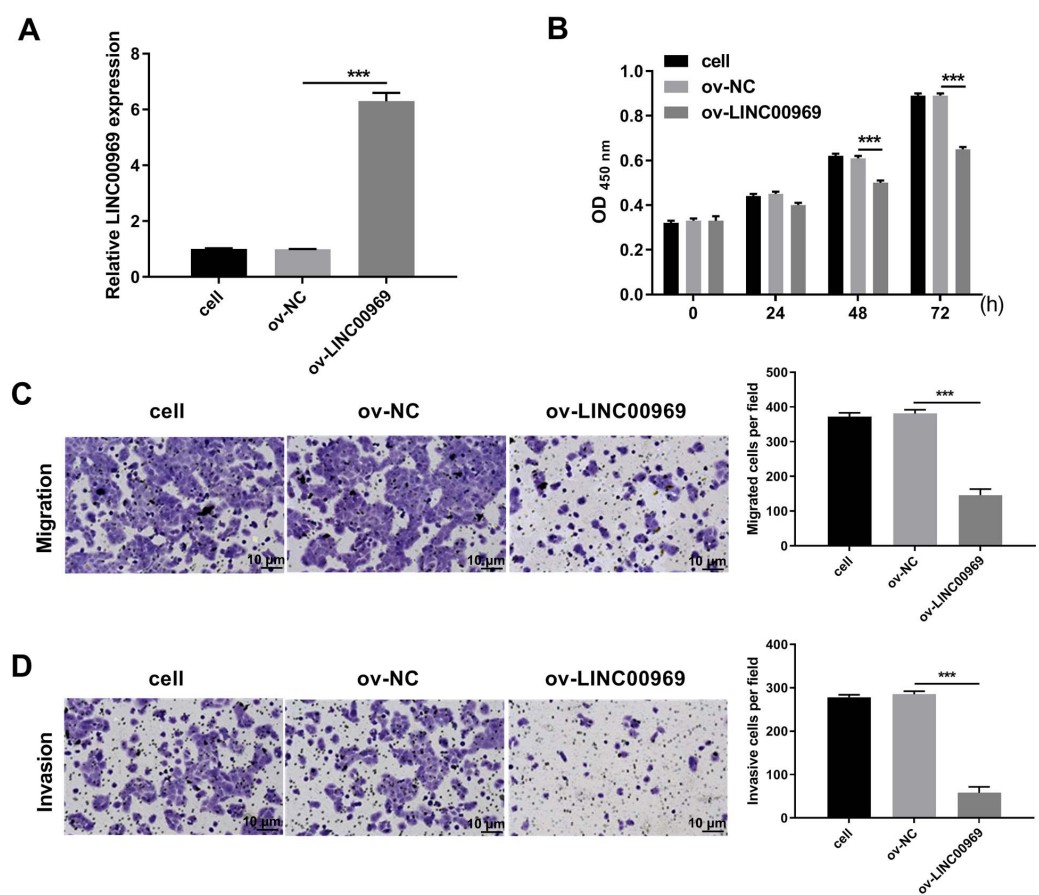

**Figure 2** **LINC00969 overexpression inhibited the malignant function of BC cells.** (A) LINC00969 expression in MCF-7 cells transfected with ov-LINC00969 or ov-NC plasmids was detected by qRT-PCR. (B) Cell proliferation of MCF-7 cells transfected with ov-LINC00969 or ov-NC plasmids was detected by CCK-8 assay. (D and E) The migration (D) and invasion (E) of MCF-7 cells transfected with ov-LINC00969 or ov-NC plasmids was assessed using the Transwell assay (magnification, 100×) (***$P$ < 0.001).

showed that HOXD8 protein expression was significantly decreased in MCF-7 cells co-transfected with si-HOXD8 and ov-LINC00969 compared to that in cells co-transfected with si-NC and ov-LINC00969 (Fig. 5A). In addition, HOXD8 knockdown enhanced PI3K and p-AKT protein expression in MCF-7 cells co-transfected with si-HOXD8 and ov-LINC00969 (Fig. 5B). HOXD8 knockdown enhanced proliferation (Fig. 5C), migration, and invasion (Figs. 5D–5E) of MCF-7 cells in the si-HOXD8+ov-LINC00969 group compared with cells in the si-NC+ov-LINC00969 group.

## LINC00969 overexpression suppresses BC cells through HOXD8/ ILP2

Our previous studies showed that HOXD8 impedes the proliferation and migration of BC cells by blocking ILP2 expression (*Wen, Chen & Fang, 2021*). Consistently, ILP2 protein expression in four BC cell lines was dramatically elevated, especially in MCF-7 cells, compared to MCF-10A cells (Fig. 6A). The markedly upregulated expression of ILP2

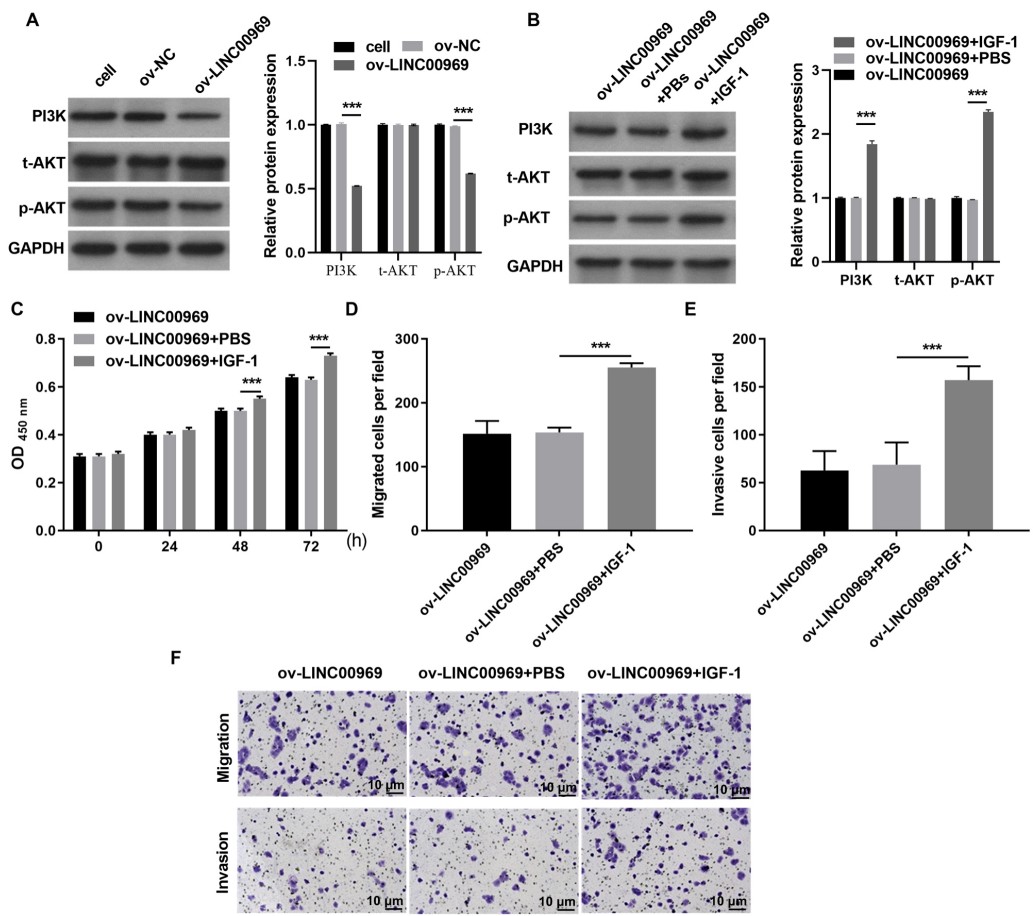

**Figure 3** **Activation of the PI3K/AKT pathway can alleviate the repressive effect of LINC00969 overexpression on MCF-7 cells.** (A) PI3K, t-AKT, and p-AKT protein expressions in MCF-7 cells transfected with ov-LINC00969 or ov-NC plasmids detected by western blot. (B) PI3K, t-AKT with p-AKT protein expression in MCF-7 cells with LINC00969 overexpression treated with IGF-1 detected by western blot. (C) Proliferation of MCF-7 cells accompanying LINC00969 overexpression after treatment with IGF-1 analyzed by the CCK-8 assay. Migration (D) and invasion (E) of MCF-7 cells overexpressing LINC00969 after treatment with IGF-1 analyzed by the Transwell assay (magnification, 100×) (***$P < 0.001$).

protein in MCF-7 cells in ov-LINC00969 group than that ILP2 protein in MCF-7 cells in ov-NC group (Fig. 6B). Notably, activation of PA-P in MCF-7 cells accompanied by LINC00969 overexpression had no effect on ILP2 protein expression (Fig. 6C). However, HOXD8 knockdown in MCF-7 cells accompanied by LINC00969 overexpression elevated ILP2 protein expression (Fig. 6D).

## LINC00969 regulated HOXD8 *via* binding to miR-425-5p

To further investigate the interaction mechanism between LINC00969 and HOXD8, we analyzed the miRNA that binds LINC00969 and HOXD8 3′UTR together. The results indicate that there are seven potential miRNAs that can simultaneously bind LINC00969 and HOXD8 3′UTR (Fig. 7A). According to the ceRNA mechanism, LINC00969 and HOXD8 are lowly expressed in BC and play an anti-tumor role. Therefore, we further

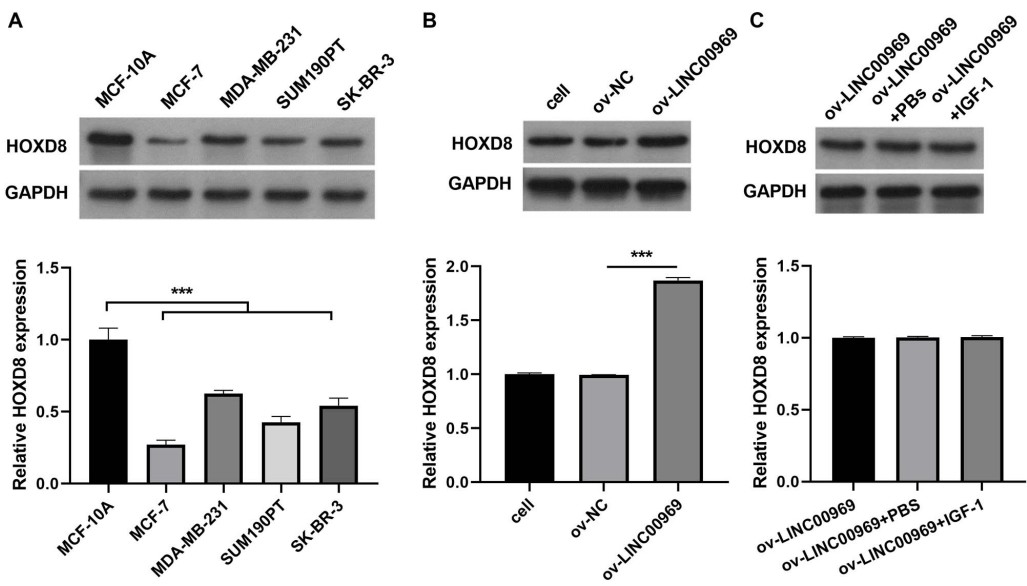

**Figure 4  HOXD8 expression is regulated by LINC00969 in BC.** (A) Western blot evaluation of HOXD8 protein expression in BC cell lines. (B and C) HOXD8 protein expression in MCF-7 cells with LINC00969 overexpression (B) or MCF-7 cells with LINC000969 overexpression treated with IGF-1 (C) assessed by western blot. (***$P < 0.001$).

selected miRNAs that are highly expressed in BC and play an oncogenic role as research objects. Through literature analysis, only miR-425-5p was found to be highly expressed in BC and play an oncogenic role (*Zhang et al., 2020*). Hence, miR-425-5p was selected for follow-up research. The luciferase experiment showed that compared with the control group, the R/F value in WT LINC00969+miR-425-5p and WT HOXD8 3′UTR +miR-425-5p groups decreased, while the R/F value in MUT LINC00969+miR-425-5p and MUT HOXD8 3′UTR +miR-425-5p groups remained unchanged (Fig. 7B). The results indicate that LINC00969 and HOXD8 3′UTR can bind to miR-425-5p. Further overexpression of miR-425-5p in MCF7 cells overexpressing LINC00969 revealed a significant upregulation of miR-425-5p expression, as well as a significant increase in proliferation, migration, and invasion of MCF7 cells (Figs. 7C–7E).

## DISCUSSION

BC is a global health concern that affects women worldwide and is associated with significant morbidity and mortality (*Kashyap et al., 2022*). The main cause of high mortality in BC is recurrence with metastasis, highlighting the need to understand the mechanisms of cancer progression and identify new therapeutic targets (*Yeeravalli & Das, 2021*). Recent studies have identified multifarious lncRNAs that are aberrantly expressed in various cancers, including BC, and play a dynamic role in gene regulation (*Reggiardo, Maroli & Kim, 2022*; *Smolarz, Zadrozna-Nowak & Romanowicz, 2021*). For instance, lncRNA ZNF582-AS1 expression was inhibited in BC tissues and its high expression was also related to a lower risk of relapse and death (*Wang et al., 2022*). LncRNA SNHG14 was remarkably elevated

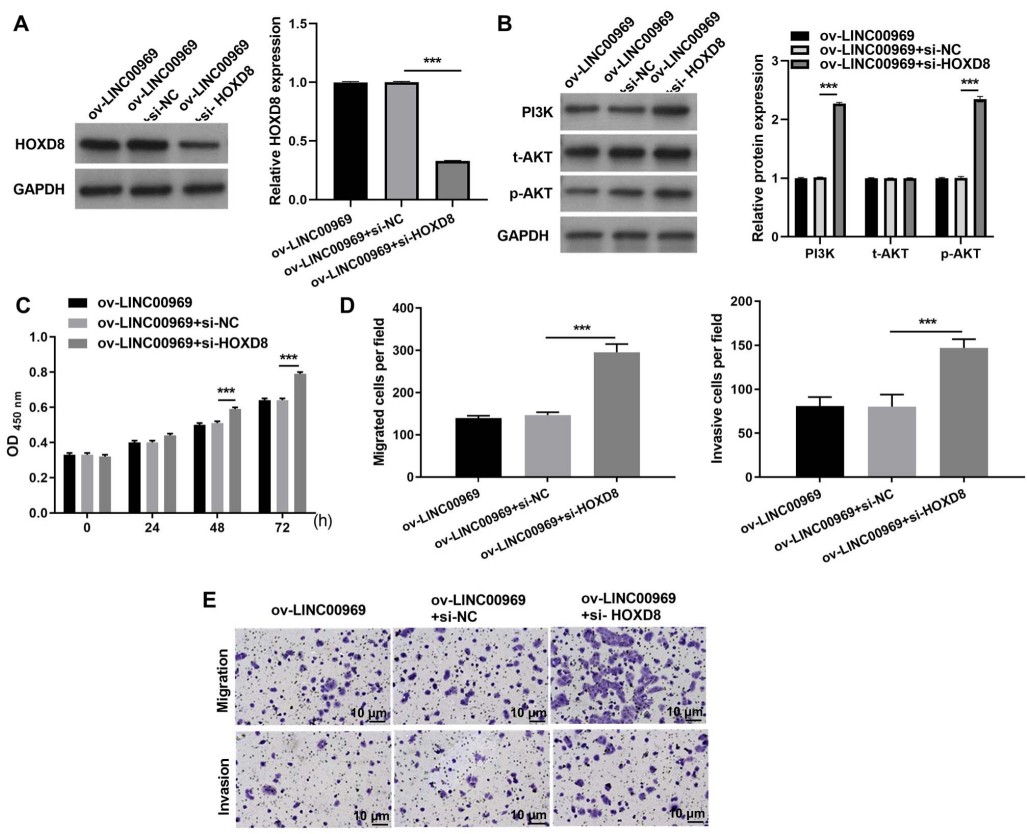

**Figure 5** **HOXD8 knockdown activates PA-P and alleviates the repressive effect of LINC00969 over-expression on BC cells.** (A and B) HOXD8, PI3K, t-AKT, and p-AKT protein expression levels in MCF-seven cells co-transfected si-HOXD8 and ov-LINC00969 were measured by western blot. (C) Proliferation of MCF-seven cells co-transfected si-HOXD8 and ov-LINC00969 analyzed by the CCK-8 assay. (D and E) Migration and invasion of MCF-seven cells co-transfected with si-HOXD8 and ov-LINC00969 were analyzed by the Transwell assay (magnification, $100\times$) (***$P < 0.001$).

and silencing SNHG14 remarkably mitigates BC cell proliferation, migration, invasion, and facilitates apoptosis (*Zhang, Ding & Peng, 2022*). Here, we observed a significant reduction in the expression of LINC00969 in BC tissues and cells. LINC00969 is a novel lncRNA which elevated in patients with IDD (*Zhao et al., 2016*). LINC00969 is highly expressed in nucleus pulposus tissues and promotes apoptosis of pulposus cells (*Yu et al., 2019*). LINC00969 expression is enhanced in lung cancer cells with acquired gefitinib resistance (*Dai et al., 2023*). Contrary to previous findings in lung cancer, our study revealed a significant reduction in the expression of LINC00969 and LINC00969 overexpression markedly suppresses BC cell proliferation, migration, and invasion. Our findings novelty provided valuable supporting LINC00969 as a anti-oncogene and promising biomarker for the clinical treatment of BC.

PA-P plays vital roles in BC cell proliferation, metastasis, and BC prognosis (*Qiu et al., 2020*; *Xiang et al., 2022*). Here, we further show that LINC00969 overexpression blocked PA-P in BC cells, and activating PA-P could reverse the repressive effect of LINC0096

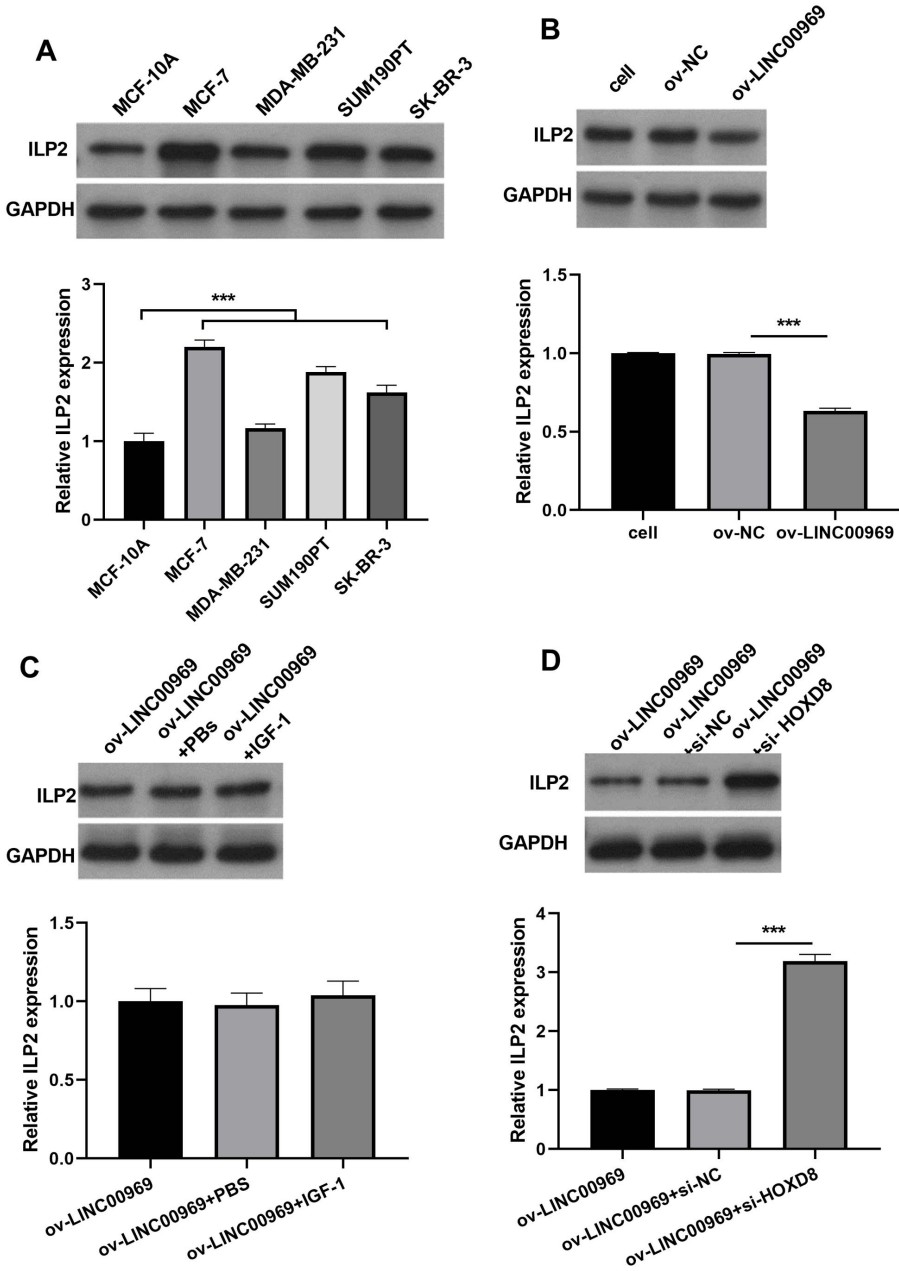

**Figure 6 LINC00969 overexpression suppresses BC cells through HOXD8/ILP2.** (A) Western blot evaluation of ILP2 protein expression in BC cell lines. (B and D) ILP2 protein expression in MCF-7 cells with LINC000969 overexpression (B), LINC000969 overexpression and co-treated with IGF-1 (C), or LINC000969 overexpression accompanied by HOXD8 knockdown (D) assessed by western blot (***$P <$ 0.001).

overexpression on the proliferation, migration, and invasion of BC cells. These results first indicate that LINC00969 overexpression inhibits the malignant behavior of BC cells by blocking PA-P.

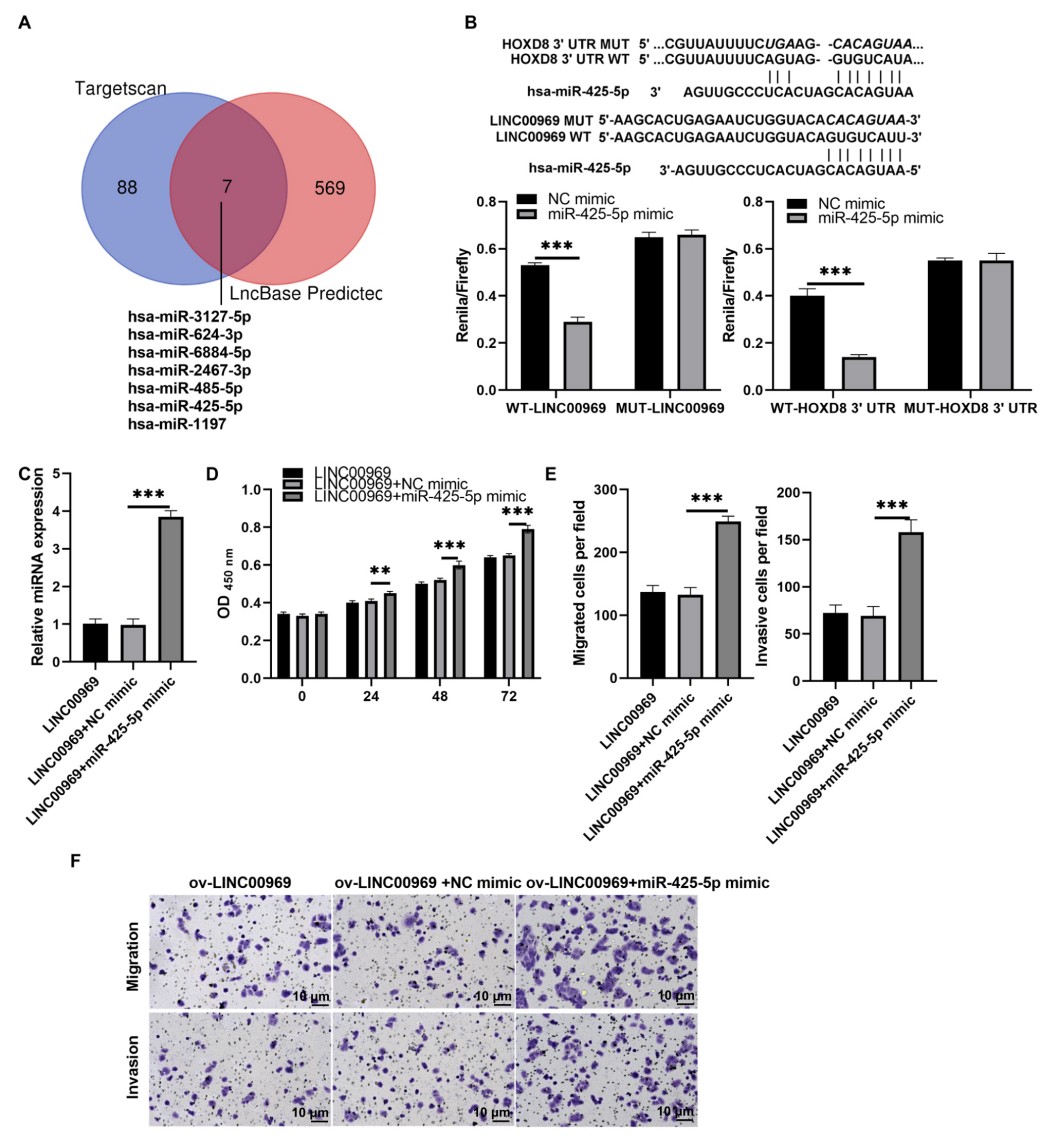

**Figure 7** **MiR-425-5p overexpression alleviates the repressive effect of LINC00969 overexpression on BC cells.** (A) The miRNAs that binds to HOXD8 3′UTR and LINC00969 were analyzed by Targetscan 8.0 and LncBase 2.0, respectively. (B) The wild type (WT) bound sites between HOXD8 3′UTR and LINC00969 and miR-425-5p were showed, and the mutational site (MUT) between HOXD8 3′UTR and LINC00969 and miR-425-5p were also showed. Then the luciferase experiment was used to analyze the bind between HOXD8 3′UTR and LINC00969 and miR-425-5p. (C) miR-425-5p expression was analyzed by qRT-PCR after co-transfection at 24 h. (D) Proliferation of MCF-7 cells co-transfected miR-425-5p mimic and ov-LINC00969 analyzed by the CCK-8 assay. (E and F) Migration and invasion of MCF-7 cells co-transfected with miR-425-5p mimic and ov-LINC00969 were analyzed by the Transwell assay (magnification, 100×). **$P < 0.01$, (***$P < 0.001$).

HOX genes are major regulators of transcription factors (*Brotto et al., 2020*). HOXD8 is a vital member of the HOX gene family, and its expression in colorectal cancer tissues is decreased, leading to the inhibition of malignant behavior in colorectal cancer cells (*Mansour & Senga, 2017*). We previously reported that HOXD8 expression was blocked in

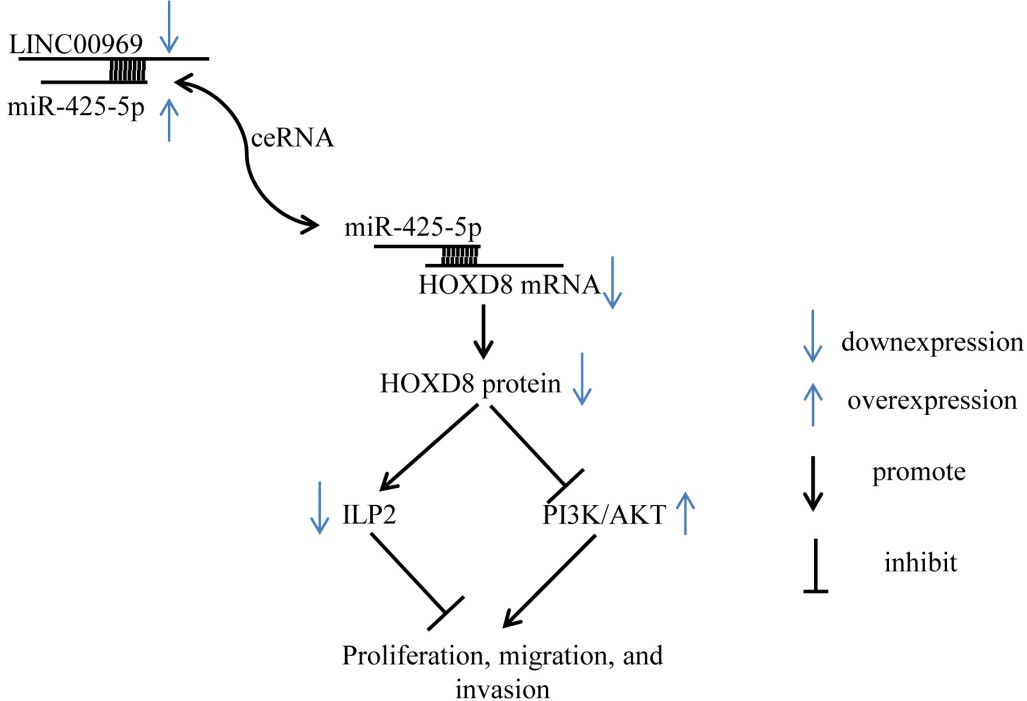

**Figure 8 Summary schematic. LINC00969 promotes the expression of HOXD8 by competitively binding miR-425-5p.** HOXD8 can promote the expression of ILP-2 and silence the PI3K/AKT signaling pathway to inhibit the proliferation, migration, and invasion of MCF7. Overall, LINC00969 inhibits the proliferation, migration, and invasion of MCF7 by competitively binding miR-425-5p to enhance HOXD8 protein, then promote ILP-2 expression and silence the PI3K/AKT signaling pathway.

BC tissues, and overexpression of HOXD8 restricted the proliferation and metastasis of cancer cells by binding to the ILP2 promoter to regulate ILP2 expression (*Wen, Chen & Fang, 2021*). Consistently, HOXD8 and ILP2 expression was regulated by LINC00969 in BC, suggesting that LINC00969 regulates the malignant behavior of BC cells *via* HOXD8/ILP2. Furthermore, previous studies have confirmed that HOXD8 is expressed at low levels in TNBC tissues and cells, and that HOXD8 overexpression could block the phosphorylation of AKT with mammalian target of rapamycin in triple-negative BC cells, suggesting that HOXD8 regulates the malignant behaviors of BC cells by blocking PA-P (*Zhang et al., 2021*). In the present study, HOXD8 expression was regulated by LINC00969 in BC, and HOXD8 knockdown activated PA-P, whereas PA-P activation had no effect on HOXD8 protein expression, indicates that PA-P is downstream of HOXD8. Importantly, HOXD8 knockdown suppressed the effect of LINC00969 overexpression in BC cells. These results first indicate that LINC00969 inhibits the proliferation and metastasis of BC by regulating the PA-P through HOXD8.

LncRNA regulates the expression of target genes by competitively binding to microRNAs. In this study, we found that LINC00969 promotes the expression of HOXD8 by competitively binding miR-425-5p. Previous study found that miR-425-5p expression was increased in BC tissues and cell lines, and is associated with poor overall survival in BC

patients. Knockout of miR-425-5p inhibited the proliferation and migration of BC cells (*Xiao et al., 2019*; *Zou et al., 2021*). Overexpression of miR-425-5p activates PA-P, while inhibition of miR-425-5p can silence PA-P (*Zhang et al., 2020*). These studies suggest that miR-425-5p acts as oncomiR. Similarly, our study indicates that miR-425-5p functions as an oncogenic miRNA, which can reverse the anticancer effect of LINC00969.It is important to note that our study has some limitations. First, the relationship between LINC00969 expression and clinical features (like DCIS, TNBC, ER+ *etc.*) of BC is not yet clear. We did not verify whether LINC00969 exhibited similar effects *in vivo* in xenografts and BC cells. Secondly, whether LINC00969 regulates HOXD8 through specific miRNAs warrants further verification. In addition, further research is needed to determine whether there is an upstream-downstream regulatory relationship between ILP-2 and PA-P. Finally, MCF7 cells are less invasive than other breast cancer cells, such as MDA-MB-231. However, the expression of LINC00969 in MCF7 cells is lower than that in MDA-MB-231 cells. The low expression of linc00969 and high invasiveness has not negatively correlated. The reasons for this need further research.

Despite these limitations, our study first demonstrates that LINC00969 is dramatically downregulated in BC and that LINC00969 inhibits the proliferation of BC cells by silencing PA-P and inhibiting ILP2 protein *via* HOXD8 (Fig. 8). Thus, LINC00969 may be a novel molecular target for BC treatment.

### Funding

The authors received no funding for this work.

### Competing Interests

The authors declare there are no competing interests.

### Author Contributions

- Xiaoyun Wen conceived and designed the experiments, performed the experiments, analyzed the data, prepared figures and/or tables, authored or reviewed drafts of the article, and approved the final draft.
- Ya Hou performed the experiments, prepared figures and/or tables, and approved the final draft.
- Liang Zhou performed the experiments, prepared figures and/or tables, and approved the final draft.
- Xiansong Fang conceived and designed the experiments, analyzed the data, authored or reviewed drafts of the article, and approved the final draft.

### Data Availability

The raw data are available at figshare: Fang, Xiansong (2023). LINC00969 inhibits the proliferation with metastasis of breast cancer by regulating the phosphorylation of

PI3K/AKT through HOXD8/ILP2. figshare. Figure. https://doi.org/10.6084/m9.figshare.23651373.v1.

## Supplemental Information

Supplemental information for this article can be found online at http://dx.doi.org/10.7717/peerj.16679#supplemental-information.

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
