# Peer review of "LINC00969 inhibits proliferation with metastasis of breast cancer by regulating phosphorylation of PI3K/AKT and ILP2 expression through HOXD8"

_PeerJ, doi:10.7717/peerj.16679_

## Round 0.1 · original submission · Major Revisions

As you can see from the comments made by the reviewers, although they affirmed the research value and scientific nature of your manuscript to a certain extent, they also put forward many comments and suggestions that deserve your serious consideration and need to be explained. Please take it seriously and respond one by one so that the quality of your manuscript can be further improved.

**Language Note:** The review process has identified that the English language must be improved. PeerJ can provide language editing services - please contact us at copyediting@peerj.com for pricing (be sure to provide your manuscript number and title). Alternatively, you should make your own arrangements to improve the language quality and provide details in your response letter. – PeerJ Staff

Reviewer 1 ·

Basic reporting

The English language should be improved to ensure that an international audience can clearly understand your text. Some examples where the language could be improved include Figure 1,2 legend.

Experimental design

3. Materials & Methods
3.1 The author should define the grouping in the Cell culture and transfer section.
3.2 line 132: The items number of the antibody should be provided.
3.3 line 146: Indicate where t-tests are used

Validity of the findings

4. Results:
4.1 line 155: “both BC cell lines”?
4.2 Figure 1 legend: LINC00969 is decreased in BC. LINC00969 expression?
4.3 Figure 1 legend: qRT-PCR evaluation of LINC000969 expression of BC cells?
4.4 line 162: Which group should be indicated for high expression.
4.5 line 165: Transwell's detection time? And the magnification should be displayed in the Figure 2 legend.
4.6 line 168: What is inhibited in BC cells?
4.7 p-AKT and t-AKT needs to be defined when it first appears。
4.8 line 174: IGF-1 treatment should be placed in the Materials and Methods section.
4.9 Figure 3B, WB image miss protein name.
4.10 line 185: both BC cell lines?
4.11 Figure 4 legend: The meaning of the *** needs to be specified.
4.12 line 198-201: Fig 6 should be fig 5.
4.13 line 210. Result description was error in Figure 6B-6D.
4.14 Figure 6 legend: Description was error in Figure 6B-6D. The meaning of the *** needs to be specified.

Additional comments

In the manuscript, the authors by combining GEPIA open access data from and In vitro experiments identified LINC00969 as a potential important target in BC progression and metastasis. The further study found that LINC00969 inhibits the proliferation and metastasis of BC cells by regulating PI3K/AKT phosphorylation through HOXD8/ILP2. This is an interesting and original study for BC, most of the figures and different panels are well constructed and convincing. Unfortunately, this article cannot be accepted in its present form. Numerous sections (both in the main text, MM and the legends) need a complete revision. Some of my remarks are listed below.
1. Abstract:
In Background, the role of LINC00969 in other tumors should be briefly described.
2. Introduction
2.1 line 62: “pivotal regulatory role”. It should be clearly stated whether it is positive regulation or negative regulation.
2.2 The role of PI3K/AKT signaling pathway in breast cancer should be described, otherwise it is impossible to clarify why PI3K/AKT signaling pathway should be studied in the downstream of LINC00969.
3. Materials & Methods
3.1 The author should define the grouping in the Cell culture and transfer section.
3.2 line 132: The items number of the antibody should be provided.
3.3 line 146: Indicate where t-tests are used
4. Results:
4.1 line 155: “both BC cell lines”?
4.2 Figure 1 legend: LINC00969 is decreased in BC. LINC00969 expression?
4.3 Figure 1 legend: qRT-PCR evaluation of LINC000969 expression of BC cells?
4.4 line 162: Which group should be indicated for high expression.
4.5 line 165: Transwell's detection time? And the magnification should be displayed in the Figure 2 legend.
4.6 line 168: What is inhibited in BC cells?
4.7 p-AKT and t-AKT needs to be defined when it first appears。
4.8 line 174: IGF-1 treatment should be placed in the Materials and Methods section.
4.9 Figure 3B, WB image miss protein name.
4.10 line 185: both BC cell lines?
4.11 Figure 4 legend: The meaning of the *** needs to be specified.
4.12 line 198-201: Fig 6 should be fig 5.
4.13 line 210. Result description was error in Figure 6B-6D.
4.14 Figure 6 legend: Description was error in Figure 6B-6D. The meaning of the *** needs to be specified.
5. Discussion
5.1 line 245: The regulatory relationship between lncRNA and PI3K/AKT in BC should be described.
5.2 line 262: It should further indicate the upstream and downstream relationship between HOX8 and PI3K/AKT
5.3 line 263: Importantly, HOXD8 knockdown suppressed LINC00969 overexpression in BC cells. What mean?
5.4 The novelty and innovative potential of your manuscript compared to the published literature should be described in more detail in the discussion section.
5.5 It should further indicate the upstream and downstream relationship between ILP2 and PI3K/AKT
6. The English language should be improved to ensure that an international audience can clearly understand your text. Some examples where the language could be improved include Figure 1,2 legend.

Reviewer 2 ·

Basic reporting

In this study the authors report that LINC00969 is significantly reduced in breast cancer tissues, and its overexpression had a profound impact on inhibiting proliferation, migration, and invasion. It also downregulated PI3K and p-AKT protein levels in MCF-7 cells. Notably, the activation of the PI3K/AKT pathway reversed the suppressive effects of LINC00969 overexpression. Additionally, LINC00969 overexpression enhanced HOXD8 and reduced ILP2 levels, but activating the PI3K/AKT pathway had no impact on HOXD8. Knocking down HOXD8 led to increased ILP2, PI3K, and p-AKT protein expression, as well as greater cell proliferation, migration, and invasion in MCF-7 cells. Overall, authors showed that LINC00969 regulates PI3K/AKT phosphorylation through HOXD8/ILP2, leading to the inhibition of proliferation and metastasis in BC cells. This study holds significant importance in the field; however, there are notable concerns outlined below that require attention.
1. There are numerous grammar and language issues, the English language and grammar needs improvement.
2. The introduction is too brief and lacks sufficient depth. It does not provide a comprehensive background on lncRNA, references only a limited number of earlier studies, and fails to establish a clear hypothesis for the current investigation.
3. If authors agree, in Figure. 1A it might be helpful to the viewers if normal is on the left and tumor is on the right side in the graph.
4. I strongly recommend that the authors refrain from using the term ‘normal’ for MCF10A, as these are not ‘normal’ primary cells but are immortalized. Some people call it ‘normal-like’. I would encourage authors to use an appropriate term.
5. When you compare multiple BC cell lines, MCF7 cells are generally known to be least invasive. But it is highly surprising that in this study compared to MDA-MB-231 cells, the LINC00969 was lesser in MCF7 cells. It would be advisable that the authors provide proof of authentication of cells as well as provide high magnification images of the cells showing morphology (control and after LINC overexpression). Authors must discuss this.
6. It will be interesting to know the if there is any correlation of LINC00969 expression with different types of breast cancer (like DCIS, TNBC, ER+ etc).
7. In lines 198 to 203 should it be Figure 5 and not Figure 6?
8. It would be highly helpful to the readers if the authors include a summary schematic.

Experimental design

NA

Validity of the findings

NA

Additional comments

NA

Reviewer 3 ·

Basic reporting

1.In this study, the researchers aimed to investigate the role of a long noncoding RNA called LINC00969 in breast cancer (BC) progression and potential regulatory mechanisms. The results showed that LINC00969 expression was significantly reduced in BC tissues. Overexpression of LINC00969 led to a marked suppression of BC cell proliferation, migration, and invasion, as well as a blockade of PI3K and p-AKT protein expression in MCF-7 cells, a commonly used BC cell line. Activation of the PI3K/AKT pathway reversed the suppressive effect of LINC00969 overexpression on BC cell proliferation, migration, and invasion. Furthermore, LINC00969 overexpression enhanced HOXD8 protein expression and blocked ILP2 protein expression in MCF-7 cells. On the other hand, activating the PI3K/AKT pathway had no effect on HOXD8 and blocked ILP2 protein expression in MCF-7 cells overexpressing LINC00969. Additionally, knockdown of HOXD8 enhanced ILP2, PI3K, and p-AKT protein expression, as well as the proliferation, migration, and invasion of MCF-7 cells co-transfected with si-HOXD8 and ov-LINC00969. Based on these findings, the researchers concluded that LINC00969 inhibits the proliferation and metastasis of BC cells by regulating PI3K/AKT phosphorylation through the HOXD8/ILP2 pathway. This suggests that LINC00969 may serve as a potential therapeutic target for BC treatment. However, the writing level of this article needs to be further improved.
2.The references are very reasonable, and the authors took the time to comb through nearly three years of research.
3.The structure of the paper is clear and the original data is reasonable.

Experimental design

4.Although the study proposes a potential mechanism by which LINC00969 regulates PI3K/AKT phosphorylation through the HOXD8/ILP2 pathway, the exact molecular mechanisms remain largely unknown cause of the selection of the key regulators needs more evidence to support. Further mechanistic studies are required to elucidate the detailed molecular interactions involved.
5.The title of the study named “LINC00969 inhibits proliferation with metastasis of breast cancer by regulating phosphorylation of PI3K/AKT through HOXD8/ILP2”. However, the research and experimental design in this paper cannot prove that the phosphorylation of PI3K/AKT was regulated through HOXD8/ILP2. Necessary experiments were needed to verify this conclusion.
6.The introduction sets the stage for the study by providing relevant background information on the importance of breast cancer, the role of lncRNAs in cancer, and the specific focus on LINC00969. It highlights the gap in knowledge regarding the role of LINC00969 in breast cancer and establishes the rationale for the study. However, the author needs to clarify the research progress of LncRNA in BC and explain why lncRNA is selected as the research object.
7.There are many pathways that have been found to affect BC, such as HER2, p53, and Wnt/β-catenin signaling, why the PI3K/AKT pathway is chosen for verification needs further clarification. In addition, why was only AKT phosphorylated protein content detected but not PI3K phosphorylation level detected?
8. The sequence of SiRNA involved in the study should be added to the article, such as si-HOXD8 and si-NC. In addition, the equipment model and reagent number used in the materials and methods should be marked, such as microplate reader, microscopy and the antibodies from Abcam.

Validity of the findings

9.These findings in Figure2 suggest that LINC00969 overexpression has the potential to inhibit cell proliferation, migration, and invasion in BC cells, highlighting its important role in suppressing the aggressive behavior of BC. More indicators need to be detected to verify the important role of LINC00969 in BC, such as apoptosis, cycle, etc.
10.The author indicated that “HOXD8 knockdown alleviates the repressive effect of LINC00969 overexpression on BC cells via PA-P”. However, the title included that “LINC00969 inhibits proliferation with metastasis of breast cancer by regulating phosphorylation of PI3K/AKT through HOXD8/ILP2”. Authors need to clarify whether HOXD8 influences PI3K/AKT or PI3K/AKT influence HOXD8.
11. Please verify whether the results in Figure 6 correspond to those in Figure 6.
12. In Discussion, the author indicated that “The findings of the present study concerning the drastically reduced LINC00969 expression in BC tissues and cells are inconsistent with previous findings.” The author needs to discuss these inconsistent results rather than just a brief description. In addition, the existing results of the previous study do not need to be discussed again.
13.The graphs of invasion and migration in the article need to add rulers, and the notes need to be described in detail, especially the ICONS of significant differences need to be marked.

---

## Round 0.2 · Minor Revisions

Although two reviewers have no questions about your revisions, one reviewer raised several minor issues with your manuscript. Please revise it as soon as possible and submit it as required.

Reviewer 1 ·

Basic reporting

Good

Experimental design

Good

Validity of the findings

Good

Additional comments

Good

Reviewer 2 ·

Basic reporting

The authors have addressed my concerns. However there is still a minor refinement required after which the manuscript can be accepted.
1. There are still several grammatical mistakes. Authors will have to pay more attention and fix it throughout the. manuscript (Some Eg: Lines 201, 240, 248, Fig. versus Figure).
2. Some labels are hidden in Figures.
3. Some labelings are very small in the figures.
4. The schematic is not easily understandable and not yet self-explanatory.

Experimental design

NA

Validity of the findings

NA

Additional comments

NA

Reviewer 3 ·

Basic reporting

The revision is effective, and the authors have done their best to improve and supplement the manuscript, which is worthy of praise.

Experimental design

no comment

Validity of the findings

no comment

Additional comments

no comment

---

## Round 0.3 · accepted · Accept

Considering that the reviewers’ comments in the previous round of review were all about the details of format and content presentation, I do not think it is necessary to send the paper for external review again after the author has made corresponding modifications. Therefore, based on the current revision status of the entire manuscript, I would make a decision to accept it.